# Tumor Tissue MIR92a and Plasma MIRs21 and 29a as Predictive Biomarkers Associated with Clinicopathological Features and Surgical Resection in a Prospective Study on Colorectal Cancer Patients

**DOI:** 10.3390/jcm9082509

**Published:** 2020-08-04

**Authors:** Masahiro Fukada, Nobuhisa Matsuhashi, Takao Takahashi, Nobuhiko Sugito, Kazuki Heishima, Yukihiro Akao, Kazuhiro Yoshida

**Affiliations:** 1Department of Surgical Oncology, Graduate School of Medicine, Gifu University, 1-1 Yanagido, Gifu City 501-1194, Japan; flyhighvb@yahoo.co.jp (M.F.); nobuhisa@gifu-u.ac.jp (N.M.); takaota@gifu-u.ac.jp (T.T.); 2United Graduate School of Drug Discovery and Medical Information Sciences, Gifu University, 1-1 Yanagido, Gifu City 501-1194, Japan; nsugito@gifu-u.ac.jp (N.S.); heishima@gifu-u.ac.jp (K.H.); yakao@gifu-u.ac.jp (Y.A.)

**Keywords:** microRNA, colorectal cancer, biomarker, MIR21, MIR29a, MIR92a, prospective study

## Abstract

Cancer-related microRNAs (miRNAs) are emerging as non-invasive biomarkers for colorectal cancer (CRC). This study aimed to analyze the correlation between the levels of tissue and plasma miRNAs and clinicopathological characteristics and surgical resection. This study was a prospective study of CRC patients who underwent surgery. Forty-four sample pairs of tissue and plasma were analyzed. The miRNA levels were evaluated by RT-qPCR. The level of tumor tissue MIR92a showed a significant difference in CRC with lymph node metastasis, stage ≥ III, and high lymphatic invasion. In preoperative plasma, there were significant differences in CRC with stage ≥ III (MIR29a) and perineural invasion (MIR21). In multivariate analysis of lymphatic invasion, the levels of both preoperative plasma MIR29a and tumor tissue MIR92a showed significant differences. Furthermore, in cases with higher plasma miRNA level, the levels of plasma MIRs21 and 29a were significantly decreased after the operation. In this study, there were significant differences in miRNAs levels with respect to the sample type, clinicopathological features, and surgical resection. The levels of tumor tissue MIR92a and preoperative plasma MIR29a may have the potential as a biomarker for prognosis. The plasma MIRs21 and 29a level has the potential to be a predictive biomarker for treatment efficacy.

## 1. Introduction

Colorectal cancer (CRC) is the third most common cancer and the second leading cause of cancer-related death in both men and women globally [1]. Therefore, for reducing the mortality associated with CRC, biomarker development associated with clinicopathological characteristics is essential in terms of planning an appropriate treatment strategy for each patient. Although it is possible to obtain more detailed information by conducting many tests before and during treatment, evaluation steps are expensive and time-consuming. Thus, alternative minimally invasive and inexpensive pretreatment tests for CRC would be helpful for patients.

One candidate examination is measuring the levels of cancer-related microRNAs (miRNAs) by liquid biopsy. MiRNAs are short (20–24 nucleotides) RNAs that are involved in the post-transcriptional regulation of gene expression in multicellular organisms, affecting both the stability and translation of mRNAs [2,3,4]. MiRNA targets protein-coding mRNA at the post-transcriptional level via direct cleavage of mRNAs or by inhibition of protein synthesis.

MiRNA biogenesis is tightly controlled, and their dysregulation is associated with carcinogenesis. Additionally, miRNAs function as signaling molecules, influencing the behavior of the recipient cells. Recent studies identified miRNAs in tissue and plasma of cancer patients, and their importance as minimally invasive liquid biomarkers for the early detection of cancer has been reported [5,6,7]. These results are important as they showed the potential of the miRNA level reflecting tumor characteristics and status in the body, but there are some limitations; almost all past studies were retrospective examinations using only a single sample (e.g., tissue sample only), the quality of the sample (e.g., frozen tissue sample without treatment with RNA stabilization reagent), and lack of detailed clinicopathological information on the patients studied (e.g., only TNM classification). In light of these limitations, it is unclear how miRNAs can be used in actual clinical situations, which ambiguity has been an obstacle to practical application.

In addition, it was unclear in those previous reports whether circulating miRNA or exosomal miRNA is appropriate for use as a biomarker. Total miRNA in the blood, called circulating miRNAs, is conventionally divided into three main categories: (i) microparticles, (ii) exosomes, and (iii) apoptotic bodies [8]. In particular, exosomal miRNA may provide the much needed layer of specificity for the development of tissue/organ-specific biomarkers; and some studies reported that it could reflect disease status and treatment response in human malignancies [9,10,11]. However, there are also some limitations including standardization of specimen handling, appropriate normalizers, isolation techniques, and low level in plasma compared to that of circulating miRNAs. Due to this, the use of the exosomal miRNA is considered unrealistic in clinical practice. Thus, circulating miRNA is expected to be used clinically as a biomarker, we decided to measure circulating plasma miRNAs levels in this study. Regarding the examined miRNAs, MIRs21, 29a, and 92a were selected as they are candidate biomarkers frequently reported in the literature [12,13,14,15,16,17,18,19].

In this study, we analyzed reliable tissue and plasma paired samples from prospectively enrolled CRC patients to clarify (i) differences in the levels of miRNAs between various samples from the same patient, and (ii) the potential of miRNAs as minimally invasive biomarker associated with the clinicopathological characteristics of CRC and surgical resection.

## 2. Experimental Section

### 2.1. Study Population

Study subjects were prospectively enrolled 44 patients at the Gifu University Hospital between July 2019 to March 2020. Case subjects consisted of patients who underwent colorectal resection of CRC (Figure 1).

The enrolled patients in this study had to meet all of the following criteria: (1) no previous history of cancer-related disease, (2) no diagnosis as synchronous double cancer, (3) did not receive radiotherapy or chemotherapy prior to surgery, (4) diagnosed as CRC pathologically following surgery, (5) no clinical diagnosis of familial adenomatous polyposis or hereditary nonpolyposis, and (6) no clinical diagnosis of inflammatory bowel disease.

The present study was conducted in accordance with the World Medical Association Declaration of Helsinki and was approved by the Ethics Committee of Gifu University (approval number 2019-074). As this study was a prospective study and included potentially identifiable patient data, informed consent was obtained from the enrolled patients. The institutional review board gave the ethics approval for this prospective study.

### 2.2. Sample Processing

All tissue sample pairs were collected during surgery, with these paired samples being from the primary colorectal tumor and its adjacent non-tumor mucosal tissue in the same patient. Non-tumor mucosal tissue was collected, taking care not to include the muscular layer, at least 5 cm away from the primary tumor. All samples were immediately stored in RNA stabilization reagent (RNA later; Qiagen, Hilden, Germany) in the operating room and kept in this reagent at 2–8 °C overnight. Then, the tissues were removed from the RNA stabilization reagent and stored at −80 °C.

Peripheral blood samples for miRNA measurement were collected from each patient the day before the operation (Pre), on postoperative day 7 (POD7), and one month postoperatively (POM1). Peripheral blood (4 mL) was added to an ethylenediamine tetra-acetic acid (EDTA)-coated anticoagulant tube, and then the plasma was isolated by centrifugation at 3000 rpm for 15 min at 4 °C. The plasma was carefully moved to a new 2.0 mL microfuge tube and stored at −80 °C until measurement of miRNA levels could be performed.

To minimize the RNA degradation, we only used samples that had been frozen/thawed only once.

### 2.3. Preparation of Exosome-Enriched Fractions

Exosome fractions were prepared by a step-wise centrifugation-untracentrifugation method. A 300 µL volume of frozen plasma samples was thawed and centrifuged for 3 min at 11,000× *g* to remove residual cell debris, and then diluted eight-fold with phosphate-buffered saline (PBS). After that, these diluted plasma samples were centrifuged at 90,000 rpm for 90 min in a TLA-110 rotor (BECKMAN COULTER, Brea, CA, USA). The pellets were resuspended in 300 µL of PBS and designated as exosome-enriched fractions. The process for extraction of exosomal miRNA from the exosome-enriched fractions was the same as that for circulating miRNA extracted from plasma, as described below.

### 2.4. Extraction of MiRNA from Tissue and Plasma Samples

MiRNAs from the frozen tissues and plasma were extracted by using a NucleoSpin^®^ microRNA kit (MACHEREY-NAGEL, Düren, Germany) and a NucleoSpin^®^ microRNA plasma kit (MACHEREY-NAGEL, Düren, Germany) according to the respective standard protocols. Tissue samples were thawed and homogenized under sterile conditions. A 300 µL volume of frozen plasma sample was thawed and centrifuged for 3 min at 11,000× *g* to remove residual cell debris. Proteins in the supernatant were precipitated by using a reagent provided in the kit and removed by centrifugation. After adjustment of the binding conditions with ethanol (tissue sample) or isopropanol (plasma sample and exosome-enriched fraction), respectively, miRNAs were bound to a miRNA collection column. To avoid contamination by cell-free tissue DNAs, a recombinant DNase in the kit was applied to digest these DNAs on the column. The miRNA was then eluted into 30 µL of RNase-free water. The yield of miRNA from tissue was checked for integrity and quality by using a microvolume spectrophotometer, and the concentration was adjusted to 10 µL/mL with RNase-free water before the RT process. The yield of circulating miRNA from plasma is usually very small, and a microvolume spectrophotometer was hardly able to detect them, unlike total miRNA from tissues. The miRNA quality was assessed by measuring circulating MIR16 levels. MIR16 is abundantly and stably found in plasma similar to ribosomal RNA among cellular RNAs and, thus, its level should reflect the degree of miRNA degradation and quality of the plasma samples.

The purified miRNA was immediately subjected to the RT process to prevent degradation of the RNA before the PCR step.

### 2.5. Reverse Transcription-Quantitative Polymerase Chain Reaction Using Real-Time Polymerase Chain Reaction

The miRNA reverse transcription-quantitative polymerase chain reaction (RT-qPCR) was performed for measuring levels of miRNAs. Total RNA was reverse-transcribed with a TaqMan^®^ Advanced miRNA cDNA Synthesis Kit (Applied Biosystems^®^, Thermo Fisher Scientific, Waltham, MA, USA). According to manufacturer’s protocols, RT-qPCR was performed with a Thermal cycler Dice Real Time System II (TaKaRa, Otsu, Japan). We performed a primer RT-PCR using TaqMan^®^ Advanced miR Assays (Applied Biosystems^®^) for hsa-MIR21-5p (Assay ID 478587_mir), hsa-MIR29a-3p (Assay ID 477975_mir), hsa-MIR92a-3p (Assay ID 477827_mir), and hsa-MIR16-5p (Assay ID 477860_mir). The sequences of these primers were as follows: 5′UAGCUUAUCAGACUGAUGUUGA3′ (hsa-MIR21-5p), 5′UAGCACCAUCUGAAAUCGGUUA3′ (hsa-MIR29a-3p), 5′UAUUGCACUUGUCCCGGCCUGU3′ (hsa-MIR92a-3p), and 5′UAGCAGCACGUAAAUAUUGGCG3′ (hsa-MIR16-5p). PCR amplifications were performed in triplicate by using a THUNDERBIRD probe qPCR Mix (Toyobo, Osaka, Japan), following the manufacturer’s procedures.

The threshold cycle (Ct) was automatically calculated by the second-derivative maximum method to minimize errors due to variation in the manual threshold determination and to differences in background fluorescence in the samples and runs. The Ct values were identified at the cycle of maximum fluorescence acceleration, the beginning point of the log-linear phase in the amplification curve.

MIR16 was selected as an internal control of plasma samples in this study because it was the best internal control available for normalizing plasma miRNAs according to our previous study [20]. For relative quantification, the –ΔCt method was performed.

### 2.6. Collection of Clinical and Pathological Characteristics

We collected information on patient characteristics including gender, age, body mass index (BMI), primary colorectal tumor location, preoperative tumor marker level, macroscopic types (Type 0: superficial type, Type 1: polypoid type, Type 2: ulcerated type with clear margin, Type 3: ulcerated type with infiltration, Type 4: diffusely infiltrating type, Type 5: unclassified type), maximum tumor diameter, pathological Union for International Cancer Control-TNM classification (8^th^ edition) (T: depth of tumor invasion, N: lymph node metastasis, M: distant metastasis) [21], histological types, lymphatic invasion (Ly: invasion of tumor cells into lymphatic vessels), venous invasion (V: invasion of tumor cells into blood vessels), and perineural invasion (Pn: invasion of tumor cells along nerve fascicles). All histological characteristics were based on the Japanese Classification of Colorectal, Appendiceal, and Anal Carcinoma: the 3rd English Edition [22]. In this study each pathological characteristics was classified into two groups according to the degree; (i) histological types: well and moderately differentiated group (tub1: well differentiated type,tub2: moderately differentiated type) vs. poorly differentiated group (por: poorly differentiated type, muc: mucinous type), (ii) Ly: mild invasion group (Ly0: no lymphatic invasion, Ly1a: minimal lymphatic invasion) vs. high invasion group (Ly1b: moderate lymphatic invasion, Ly1c: severe lymphatic invasion), (iii) V: mild invasion group (V0: no venous invasion, V1a: minimal venous invasion) vs. high invasion group (V1b: moderate venous invasion, V1c: severe venous invasion), and iv) Pn: negative (Pn0: no perineural invasion) vs. positive (Pn1: presence of perineural invasion).

### 2.7. Statistical Analysis

For comparisons of variables between groups, Student’s *t*-test and ANOVA followed by the Turkey–Kramer test was used in independent cases, and paired *t*-test was used in paired cases for continuous variables. Pearson’s rank correlation coefficient was used for the correlation between two continuous variables. A logistic regression model was used to estimate odds ratio (OR) and 95% confidence interval (CI) for multivariate analysis. A *p*-value less than 0.05 was considered significant. All statistical analyses were performed by using JMP software (SAS Institute Inc., Cary, NC, USA).

## 3. Results

### 3.1. Patient Characteristics

Patient characteristics are presented in Table 1. The cohort consisted of 17 male (38.6%) and 27 female (61.4%) patients. Their age ranged from 42 to 85 years, with a median of 72 years. The primary tumor location was the colon in 27 cases (61.4%) and in the rectum in 17 cases (38.6%), and on the right side in 17 cases (38.6%) and on the left side in 27 cases (61.4%). The pathological stage was stage I in nine cases (20.5%), stage II in 13 cases (29.5%), stage III in 17 cases (38.6%), and stage IV in five cases (11.4%). Compared with the cancer statistics in Japan published in 2019 [23], the enrolled patient population had a slightly higher proportion of females, but the distributions of age, tumor location, and pathological stage (surgical cases only) were generally in agreement.

### 3.2. Evaluation of the Relative Levels of Exosomal MiRNAs to the Circulating MiRNAs in Plasma

We evaluated the relative levels of exosomal miRNAs to circulating miRNAs in four preoperative plasma samples. The results showed that the relative levels of all exosomal miRNAs (MIRs16, 21, 29a, and 92a) were very small, being less than 0.05 (Table 2 and Appendix A). Therefore, in terms of stable level in plasma and ease of extraction and measurement, we decided to select circulating miRNA in the plasma as an evaluation target according to the initial plan.

### 3.3. Evaluation of the Potential Internal Control for Quantification of Tumor Tissue MiRNA

To reliably select an internal control for quantification of tissue miRNA, we examined the levels of MIRs16, 186, and 361 by using RT-qPCR for 12 pairs of tissue samples. Both MIRs186 and 361 were recommended as the candidate internal control miRNAs for use in Taqman^®^ Advanced Assays (Applied Biosystems^®^, Thermo Fisher Scientific, Waltham, MA, USA). No significant differences were identified in terms of the level of MIR16 (*p* = 0.76), MIR186 (*p* = 0.86), and MIR361 (*p* = 0.23) between tumor and normal mucosa (paired *t*-test, Appendix A). Next, we compared the abundance of MIRs16, 186, and 361 in tissue and found that the MIR16 level was significantly the highest among them (ANOVA followed by the Turkey Kramer test, Appendix A). Therefore, MIR16 was selected as the normalization internal control for tissue miRNA, the same as for plasma miRNA.

### 3.4. Levels of MIRs21, 29a, and 92a Evaluated by RT-qPCR in Colorectal Tumor Tissues and the Relationship with Clinicopathological Characteristics

We firstly analyzed the levels of MIRs21, 29a, and 92a by performing RT- qPCR on 44 pairs of tissue samples from tumor and normal mucosa. We found that the levels of MIRs21, 29a, and 92a in the tumor tissue were significantly higher than those in the normal mucosa (MIR21: *p* < 0.001, MIR29a: *p* < 0.001, and MIR92a: *p* < 0.001, respectively; Figure 2 and Appendix A).

Based on these results, we next focused on the relationship between the level of miRNAs in the tumor tissues and the clinicopathological characteristics of CRC patients (Table 3). The level of tumor tissue MIR92a was significantly decreased in CRC patients who were positive for lymph-node metastasis (*p* = 0.01), who showed a pathological Stage III + Ⅳ (*p* = 0.04), a poorly-differentiated tumor (*p* = 0.03), and high lymphatic invasion (*p* = 0.02). Thus, tumor tissue MIR92a showed a significant relationship between its level and the clinicopathological characteristics (Figure 3).

### 3.5. Correlation of MiRNA Levels Between Tumor Tissue and Preoperative Plasma

We analyzed the correlation for the miRNA levels between tumor tissue and preoperative plasma in CRC patients. However, there was no significant correlation between them for each miRNA (MIR21: *p* = 0.94, MIR29a: *p* = 0.17, MIR92a: *p* = 0.88; Figure 4). In particular, the relationship between tumor tissue and plasma levels of MIR92a may be the weakest among these miRNAs.

### 3.6. Relationship between Levels of MiRNAs in Preoperative Plasma and Clinicopathological Characteristics

We next analyzed the relationship between levels of miRNAs in preoperative plasma and clinicopathological characteristics of CRC patients, the same as we had done for tumor tissue (Table 4). The level of preoperative plasma MIR21 was significantly increased in CRC patients with perineural invasion (*p* = 0.01). On the other hand, the level of preoperative plasma MIR29a was significantly increased in CRC patients with large tumor (≥45 mm) (*p* = 0.046), and a pathological Stage III + IV (*p* = 0.02). Furthermore, the levels of both preoperative plasma MIRs21 and 29a showed a very significant tendency in CRC patients with high lymphatic invasion (*p* = 0.05, both; Figure 5). Thus, in contrast to the case of tumor tissue, these 2 miRNAs, excluding MIR92a, showed some significant relationships between their levels in preoperative plasma and the clinicopathological characteristics.

### 3.7. Multivariate Analysis of MiRNAs and Pathological Characteristics in the CRC with High Lymphatic Invasion

These results indicate that the levels of miRNAs may be associated with tumor characteristics, especially as regards lymphatic invasion. A multivariate analysis of six factors (three pathological factors: pathological T stage, venous invasion, perineural invasion, and three levels of miRNAs: tumor tissue MIR92a, preoperative plasma MIRs21 and 29a) was then carried out to identify the independent factors affecting lymphatic invasion. The levels of miRNAs were classified into high and low groups based on the median. As a result, venous invasion (*p* < 0.0001, OR = 66.84, 95% CI = 5.10–2853.98), the level of tumor tissue MIR92a (*p* = 0.01, OR = 0.08, 95%CI = 0.003–0.60), and the level of preoperative plasma MIR29a (*p* = 0.009, OR = 30.72, 95%CI = 2.24–1038.11) were confirmed to be independent factors of lymphatic invasion in CRC patients (Table 5).

### 3.8. Changes in Level of Plasma MIR21, 29a, and 92a Between Pre- and Post-Surgical Resection

The levels of MIRs21, 29a, and 92a were analyzed in paired pre-and post-operative plasma samples. The plasma levels of MIRs21, 29a, and 92a of all 36 patients (excluding eight non-curative resection cases) did not show any significant difference between pre- and post-operative samples (MIR21: Pre and POD7; *p* = 0.32, Pre and POM1; *p* = 0.83, MIR29a: Pre and POD7; *p* = 0.25, Pre and POM1; *p* = 0.62, MIR92a: Pre and POD7; *p* = 0.42, Pre and POM1; *p* = 0.68 respectively; Figure 6a). There was, thus, no clear correlation between surgical resection and changes in the levels of plasma miRNAs tested in CRC patients who had undergone curative resection.

However, there was very close relationship between the level of plasma miRNA (classified into high and low groups based on the median) and change in the level of plasma MIR21 and 29a before and one month after surgery (*p* < 0.001, *p* = 0.002, respectively; Figure 6b). Then, limited to CRC patients with high plasma miRNA level, the levels of both postoperative plasma MIRs21 and 29a showed a significant decrease compared to the preoperative level (MIR21: Pre and POD7: *p* = 0.003, Pre and POM1: *p* = 0.03, MIR29a: Pre and POD7; *p* = 0.002, Pre and POM1; *p* = 0.046; Figure 6c). On the other hand, there was no significant difference in the CRC patients with low plasma miRNA level (MIR21: Pre and POD7: *p* = 0.94, Pre and POM1: *p* = 0.83, MIR29a: Pre and POD7; *p* = 0.93, Pre and POM1; *p* = 0.99; Figure 6d). A similar analysis was conducted for MIR92a, but significant difference was found only one week after surgery, and not one month after surgery in the CRC patients with high plasma miRNA level (Figure 6c,d).

## 4. Discussion

In this study, we analyzed 44 tumor tissue and plasma paired samples from prospectively enrolled CRC patients, and found significant differences between the levels of miRNAs, sample types, and clinicopathological features. Especially, the levels of tumor tissue MIR92a and preoperative plasma MIR29a could discriminate CRC with high lymphatic invasion. Lymphatic invasion is a traditional factor used in estimating the aggressiveness of CRC [24,25,26]. Akagi et al. [27] revealed that lymphatic invasion was significantly associated with not only lymph-node metastasis but also recurrence-free survival and overall survival, based on their prospective study including 1,616 patients with a median follow-up period of 100 months. Furthermore, in cases with high plasma MIR levels, the levels of both plasma MIRs21 and 29a may rapidly reflect the residual amount of cancer cells in the body after surgical resection. Thus, these miRNAs are likely to be noninvasive biomarkers reflecting cancer progress, treatment efficacy, recurrence, and prognosis more than just a diagnosis of CRC. To our knowledge, this is the first prospective study to collect five samples (tissue (tumor/normal mucosa) and plasma (Pre/POD7/POM1)) from the same patient and to analyze relationship between the levels of miRNAs and clinicopathological characteristics and surgical resection.

Firstly, previous reports demonstrated that the level of MIR92a is significantly up-regulated in CRC and may present a novel screening biomarker for the early diagnosis of CRC [18,19]. Although the target and precise role of MIR92a in the pathogenesis of CRC is still contentious, dysregulation of MIR92a level has been detected in various cancers and is likely to be correlated with the biological mechanism of tumor development [28,29,30,31]. MIR92a belongs to the MIR17-92 cluster and is located on chromosome 13q13. MIR17-92 is known as one of the representative cancer-related miRNAs [32,33]. Tsuchida et al. [34] reported that MIR92a plays an oncogenic role in colon cancer as a key component of the MIR17-92 cluster. They found that MIR92a induces apoptosis and directly targets the anti-apoptotic molecule BCL-2 interacting mediator of cell death (BIM) in CRC tissues. Huiqing et al. [35] reported that MIR92a may promote the proliferation and migration of CRC cells through targeting Krüppel-like factor 4 (KLF4) as well as downstream p21, identified as a negative regulator in cell-cycle progression. In our previous study [36], we found that extracellular MIR92a packed within microvesicles secreted by CRC cells is delivered into endothelial cells and contributes to the proliferation and motility of these cells through down-regulation of *Dikkopf-3* (*Dkk-3*), a presumed tumor suppressor gene. Furthermore, we found the level of tumor tissue MIR92a is lower in the larger tumors (<40 mm vs. ≥40 mm; *p* = 0.017). In this study, although up-regulation of MIR92a in the tumor compared with its level in the normal mucosa was observed, the levels of tumor tissue MIR92a were also lower in the CRC with lymph node metastasis and high lymphatic invasion. We also found that tumor tissue MIR92a may not affect the level in the plasma. Thus, the level of MIR92a in tumor tissue but not in plasma may be suitable for use as a prognostic biomarker.

Secondly, MIR21 is located at the 10th intron of the coding region for transmembrane protein 49 (TMEM49) at chromosome 17q23; and it is one of the most prominent miRNAs implicated in the carcinogenesis and progression of human malignancy [37]. Its level is notably up-regulated in many cancers, including CRC, lung cancer, glioblastoma, hepatocellular carcinoma, and gastric cancer. In addition, its elevated level has been causally associated with tumor aggressiveness and poor prognosis [13,14,15,38,39,40,41,42,43].

MIR21′s targets include phosphatase and tensin homolog (PTEN), programmed cell death 4 (PDCD4), methylthioadenosine phosphorylase (MTAP), and Transforming Growth Factor-β (TGF-β), all of which regulate the cell cycle. Numerous reports have demonstrated that MIR21 acts as a potential oncogene in CRC by promoting tumorigenesis, invasion, and metastasis through regulation of these genes.

Wu et al. [44] reported that the level of PTEN protein in CRC tissues and cells is inversely correlated with MIR21 expression. The PTEN protein was reported to be a tumor suppressor gene acting by inhibiting the phosphoinositide-3 kinase (PI3K)/protein kinase B (Akt) pathway. They suggested that MIR21 modulates malignant phenotypes such as proliferation, anti-apoptosis, cell-cycle progression, and invasion in CRC cells by down-regulating PTEN protein expression. In our study, the level of preoperative plasma MIR21 was significantly increased in CRC patients with lymphatic and perineural invasion. This result is consistent with the function of MIR21 as a cancer-related miRNA, and the level of preoperative plasma MIR21 may serve as a potential prognostic biomarker reflecting cancer micro invasion.

We also analyzed the relationship between the levels of miRNAs and surgical resection. In our previous study on canine hemangiosarcoma [20], we showed that tumor-related circulating miRNAs are significantly elevated in plasma and decrease after surgical resection. From this result, we proposed that the level of tumor-related circulating miRNAs has the potential to be used as a biomarker reflecting the amount of tumor in the body, that is, as a prediction biomarker of treatment efficacy and recurrence. Juan et al. [45] also analyzed how cancer treatments affect the level of plasma MIR21 in non-small cell lung cancer. They showed that the level of MIR21 significantly decreases after chemotherapy, being more significantly reduced in the effectively treated group than in the non-effectively treated one. In contrast, after surgical resection, the level of plasma MIR21 remains high. They suggested that surgery can decrease the plasma MIR21 level, but not bring it back to normal. Similar results were obtained in our study, but a significant decrease in the plasma level of MIR21 was observed in CRC patients with high plasma MIR21 level. From this result and past reports we speculate as one hypothesis that it is possible to accurately detect a change in the miRNA level due to surgery if the target miRNA is sufficiently expressed in the preoperative plasma. In conclusion, MIR21 could be a sensitive biomarker reflecting tumor status.

Finally, MIR29a is processed from an intron of a long non-coding transcript from chromosome 7. Like the other two miRNAs, previous reports also suggested that MIR29a is frequently up-regulated in CRC tissues, implying that MIR29a is a potential cancer-promoting miRNA in CRC [46]; and the level of MIR29a has been proposed as a novel biomarker for early and noninvasive diagnosis of CRC [16,17]. In this study, we found that plasma MIR29a had significant relationships with both clinicopathological characteristics (especially lymphatic invasion) of CRC and surgical resection. Tang et al. [47] reported that an increased level of MIR29a promotes CRC metastasis by regulating matrix metalloproteinase 2 (MMP2)/E-cadherin through direct targeting of KLF4. Yuan et al. [48] showed that a decrease in the level of MIR29a can elevate PTEN expression, suppress CRC cell proliferation, and facilitate cell apoptosis. As mentioned above, PTEN and KLF4 are also reported to be target genes of MIRs21 and 92a [31,35,40,44]. Thus, the levels of different miRNAs may be correlated with each other and have common characteristics.

Interestingly, sample type (tissue or plasma), changes in levels of miRNAs (increase or decrease), and changes after surgical resection (present or absent) differ depending on the kind of miRNAs. We would like to propose that it is necessary to identify the appropriate sample type, appropriate target patients, and appropriate purpose of the biomarker (early detection, treatment efficacy, recurrence, or prognosis) according to each miRNA for practical clinical use.

Some limitations of this study need to be addressed. First, as the sample size was small, requiring further validation of these markers by using larger samples. Second, because the interval between pre- and post-operation was as short as only a month, the correlation between (i) the changes in levels of miRNAs and treatment effect and recurrence, and (ii) the preoperative levels of miRNA and prognosis is still unclear. Therefore, we are planning to increase the number of enrolled patients in this study and to continue to measure the levels of miRNAs in plasma samples and to investigate prognosis in our on-going studies. In the future, by conducting large-scale prospective research in multi institutions, it will be necessary to build a biomarker system that enables more accurate understanding of the tumor status in CRC patients.

## 5. Conclusions

Our results indicate that the levels of MIRs21, 29a, and 92a showed the significant differences in sample types, clinicopathological characters of CRC, and surgical resection. In particular, the levels of tumor tissue MIR92a and preoperative plasma MIR29a showed a significant correlation with high lymphatic invasion. Thus, they have the potential as a biomarker for prognosis. The levels of plasma MIRs21 and 29a significantly decreased after surgical resection, limited to CRC patients with high plasma miRNA level. In addition, the level of plasma MIRs21 and 29a may be biomarker used for treatment efficacy.

## Figures and Tables

**Figure 1 jcm-09-02509-f001:**
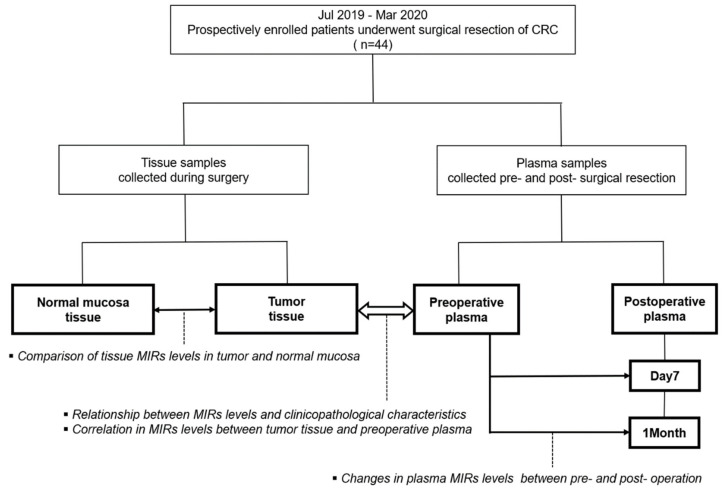
Study flow chart.

**Figure 2 jcm-09-02509-f002:**
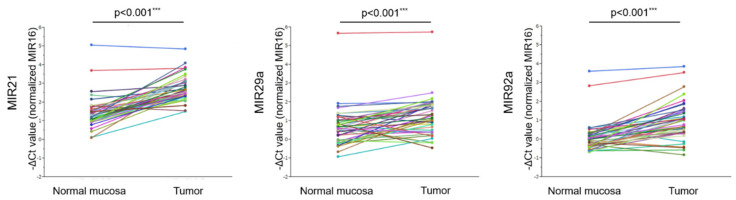
Levels of MIRs21, 29a, and 92a in tumor and normal mucosa tissues of 44 CRC patients. The paired *t*-test was performed. (***: *p* < 0.001).

**Figure 3 jcm-09-02509-f003:**
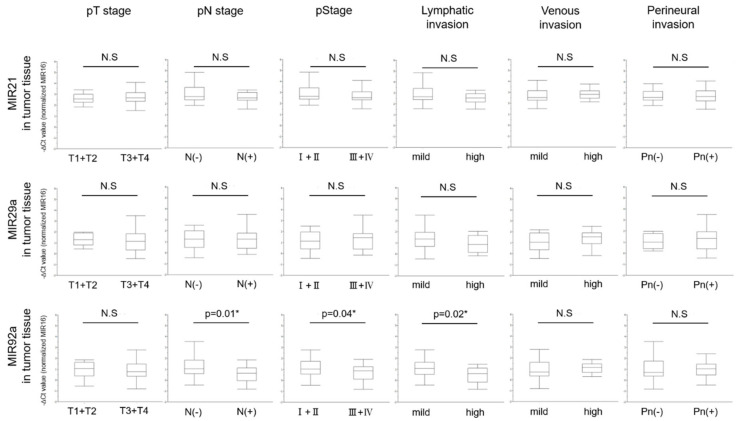
Relationship between levels of MIRs21, 29a, and 92a in tumor tissue and clinicopathological characteristics of 44 CRC patients. Student-t test was performed. (*: *p* < 0.05).

**Figure 4 jcm-09-02509-f004:**
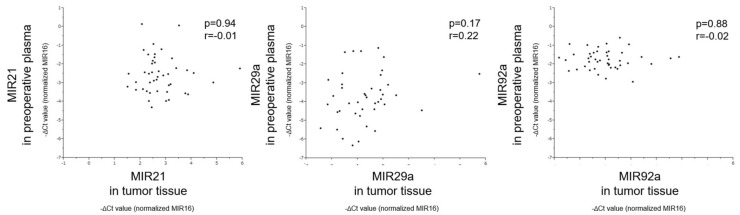
Correlation between the levels of MIRs21, 29a, and 92a in preoperative plasma and in tumor tissue of 44 CRC patients. Pearson’s rank correlation coefficient (r) is shown.

**Figure 5 jcm-09-02509-f005:**
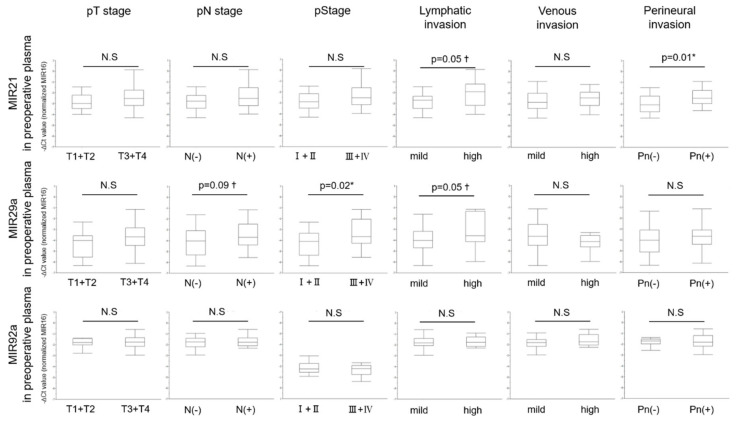
Relationship between levels of MIRs21, 29a, and 92a in preoperative plasma and clinicopathological characteristics of 44 CRC patients. Student’s *t*-test was performed. (†: *p* < 0.1 *: *p* < 0.05).

**Figure 6 jcm-09-02509-f006:**
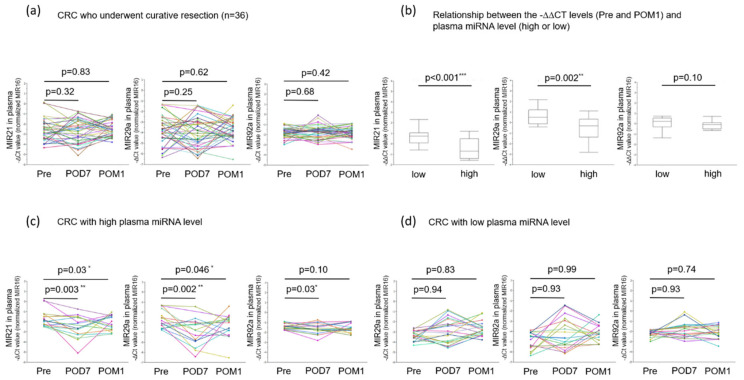
(**a**) Levels of MIRs21, 29a, and 92a preoperatively and at POD7 and POM1 of 36 CRC patients who underwent curative resection. The paired *t*-test was performed for comparisons between pre–and post-operative groups. (**b**) Relationship between the −∆∆CT levels (before operation and one month after operation) of MIRs21, 29a, and 92a in plasma of 36 CRC patients and plasma MIR level. Student’s *t*-test was performed for comparisons between groups. (**c**) Levels of MIRs21, 29a, and 92a preoperatively and at POD7 and POM1 of CRC patients with high plasma MIR level. The paired *t*-test was performed for comparisons between pre- and post-operative groups. (**d**) Levels of MIRs21, 29a, and 92a preoperatively and at POD7 and POM1 of CRC patients with low plasma MIR level. The paired *t*-test was performed for comparisons between pre- and post-operative groups (Pre: the day before the operation, POD7: on postoperative day 7, POM1: on postoperative one month).

**Table 1 jcm-09-02509-t001:** Patients’ and clinicopathological characteristics.

	Patient Total *n* = 44
Gender, *n* (%)	Male 17 (38.6) Female 27 (61.4)
Age (years), median (range)	72 (42–85)
BMI, median (range)	22.1 (15.1–40.3)
Primary colorectal tumor location, *n* (%)	C 3 (6.8)/A 8(18.2)/T 6 (13.6)/D 2 (4.5)/S 8 (18.2)/R 17 (38.6)
	Colon 27 (61.4)/Rectum 17 (38.6)
	Right side 17 (38.6)/Left side 27 (61.4)
Macroscopic types^§^, *n* (%)	Type0 4 (9.1)/Type1 4 (9.1)/Type2 29 (65.9)/Type5 7 (15.9)
Maximum tumor diameter (mm), median (range)	45 (15–180)
Pathological T Stage^§^, *n* (%)	T1 4 (9.1)/T2 7 (15.9)/T3 21 (47.7)/T4 12 (27.3)
Pathological N Stage^§^, *n* (%)	N0 24 (54.6)/N1 14 (31.8)/N2 6 (13.6)
Pathological Stage^§^, *n* (%)	I 9 (20.5)/II 13 (29.5)/III 17 (38.6)/IV 5 (11.4)
Histological types^¶^, *n* (%)	tub1 9 (20.5)/tub2 29 (65.9)/por 1 (2.3)/muc 5 (11.4)
Lymphatic invasion^¶^, *n* (%)	Ly0 8 (18.2)/Ly1a 23 (52.3)/Ly1b 8(18.2)/Ly1c 5 (11.4)
Venous invasion^¶^, *n* (%)	V0 12 (27.2)/V1a 17 (38.6)/V1b 8 (18.2)/V1c 7 (15.9)
Perineural invasion^¶^, *n* (%)	Pn0 19 (43.2)/Pn1 25 (56.8)
Preoperative CEA, *n* (%)	Normal 26 (59.1) Elevated 18 (40.1)
Preoperative CA19-9, *n* (%)	Normal 32 (72.7) Elevated 12 (27.3)

C: Cecum, A: Ascending colon, T: Transverse colon, D: Descending colon, S: Sigmoid colon, R: Rectum, Right side: Cecum, ascending colon, and transverse colon, Left side: Descending colon, sigmoid colon, and rectum, CEA: Carcinoembryonic antigen level, normal upper limit at 5 ng/mL, CA19-9: Carbohydrate antigen 19-9 level, normal upper limit at 37 ng/mL, §: UICC TNM classification (the 8th edition); ¶: the Japanese Classification of Colorectal, Appendiceal, and Anal Carcinoma: the 3rd English Edition.

**Table 2 jcm-09-02509-t002:** Expression levels of exosomal MIR16, 21, 29a, and 92a relative to those of circulating microRNAs in plasma.

	Target MicroRNAs			
	MIR16	MIR21	MIR29a	MIR92a
−ΔCt value ^‡^, median (range)	−6.35 (−9.72–−5.10)	−4.81 (−6.86–−3.72)	−5.11 (−7.10–−4.50)	−6.35 (−9.72–−5.1)
Relative expression level ^§^, median (range)	0.014 (0.001–0.029)	0.038 (0.009–0.076)	0.029 (0.007–0.12)	0.036 (0.014–0.044)

‡: −ΔCt = −(Ct value (exosomal target MIR)–Ct value (circulating target MIR)), §: calculated by 2^−ΔCt^ method.

**Table 3 jcm-09-02509-t003:** Relationship between the levels of microRNAs in tumor tissue and clinicopathological characteristics.

			Target MicroRNAs					
			MIR21		MIR29a		MIR92a	
			−∆Ct value ^‡^	*p*-value	−∆Ct value^‡^	*p*-value	−∆Ct value ^‡^	*p*-value
Gender	Male	17 (38.6)	2.58 (1.51–5.91)	0.87	0.92 (−0.16–3.5)	0.25	1.04 (−0.56–2.07)	0.41
	Female	27 (61.4)	2.57 (1.55–4.86)		1.36 (−0.44–5.75)		1.01 (−0.82–3.87)	
Age	≥75	14 (31.8)	2.30 (1.51–5.91)	0.66	1.42 (−0.44–5.75)	0.54	0.75 (−0.44–3.55)	0.82
	<75	30 (68.2)	2.61 (1.55–4.86)		1.17 (−0.14–2.50)		1.04 (−0.82–3.87)	
BMI	≥22	22 (50.0)	2.48 (1.55–3.87)	0.03^*^	0.4 (−0.17–2.50)	0.09†	0.75 (−0.82–2.41)	0.28
	<22	22 (50.0)	2.96 (1.51–5.91)		1.62 (−0.44–5.75)		1.10 (−0.44–3.87)	
Tumor location	Colon	27 (61.4)	2.67 (1.51–5.91)	0.70	1.14 (−0.44–3.5)	0.47	0.71 (−0.82–2.79)	0.25
	Rectum	17 (38.6)	2.53 (1.55–4.86)		1.32 (−0.16–5.75)		1.12 (−0.56–3.87)	
	Right side	17 (38.6)	2.52 (1.51–4.1)	0.16	0.84 (−0.44–2.19)	0.06†	0.68 (−0.82–2.41)	0.11
	Left side	27 (61.4)	2.65 (1.55–5.91)		1.36 (−0.16–5.75)		1.12 (−0.56–3.87)	
Preoperative CEA	Elevated	18 (40.1)	2.57 (1.55–5.91)	0.87	1.21 (−0.44–5.75)	0.64	1.04 (−0.82–3.55)	0.57
	Normal	26 (59.1)	2.61 (1.51–4.86)		1.27 (−0.17–2.5)		0.86 (−0.56–3.87)	
Preoperative CA19-9	Elevated	12 (27.3)	2.54 (1.55–3.13)	0.14	1.01 (−0.16–2.05)	0.21	1.10 (−0.82–1.56)	0.07 †
	Normal	32 (72.7)	2.62 (1.51–5.91)		1.36 (−0.44–5.75)		1.08 (−0.56–3.78)	
Maximum tumor diameter	≥45	22 (50.0)	2.57 (1.51–5.91)	0.95	1.34 (−0.17–3.5)	0.97	0.75 (−0.56–2.07)	0.16
	<45	22 (50.0)	2.65 (1.84–4.86)		1.04 (−0.44–5.75)		1.07 (−0.82–3.87)	
Pathological T stage ^§^	T3 + T4	33 (75.0)	2.67 (1.51–5.91)	0.77	1.14 (−0.44–5.75)	0.79	0.78 (−0.82–3.55)	0.51
	T1 + T2	11 (25.0)	2.57 (1.84–4.86)		1.28 (0.43–1.99)		1.07 (−0.56–3.87)	
Pathological N stage ^§^	Present	20 (45.4)	2.53 (1.51–5.91)	0.35	1.23 (−0.16–3.5)	0.57	0.64 (−0.82–1.87)	0.01 *
	Absent	24 (54.5)	2.67 (1.84–4.86)		1.24 (−0.44–5.75)		1.06 (−0.44–3.87)	
Pathological Stage ^§^	III + IV	22 (50.0)	2.53 (1.51- 5.91)	0.55	1.44 (−0.16–3.50)	0.85	0.86 (−0.82–1.92)	0.04 *
	I + II	22 (50.0)	2.665 (1.84–4.86)		1.12 (−0.44–5.75)		1.03 (−0.44–3.87)	
Histological differentiation ^¶^	Poorly	6 (13.6)	2.23 (1.51–2.83)	0.03*	0.6 (−0.16–1.92)	0.16	0.13 (−0.56–1.38)	0.03 *
	Well and moderately	38 (86.4)	2.66 (1.84–5.91)		1.32 (−0.44–5.75)		1.06 (−0.82–3.87)	
Lymphatic invasion ^¶^	High	13 (29.5)	2.53 (1.51–3.24)	0.14	0.84 (−0.17–2.05)	0.13	0.6 (−0.82–1.46)	0.02 *
	Mild	31 (70.4)	2.65 (1.55–5.91)		1.35 (−0.44–5.75)		1.08 (−0.44–3.87)	
Venous invasion ^¶^	High	15 (34.1)	2.83 (2.14–5.91)	0.29	1.55 (−0.17–3.5)	0.33	1.12 (−0.56–1.87)	0.97
	Mild	29(65.9)	2.52 (1.51–4.86)		1.04 (−0.44–5.75)		0.71 (−0.82–3.87)	
Perineural invasion ^¶^	Positive	25 (56.8)	2.68 (1.51–5.91)	0.97	1.35 (−0.44–3.5)	0.97	1.04 (−0.44–2.41)	0.61
	Negative	19 (43.2)	2.58 (1.84–4.86)		1.01 (0.2–5.75)		0.71(−0.82–3.87)	

CEA: Carcinoembryonic antigen level, normal upper limit at 5 ng/mL, CA19-9: Carbohydrate antigen 19-9 level, normal upper limit at 37 ng/mL, ‡: −∆Ct value = −(Ct value (target MIR)–Ct value (internal control MIR16)), §: UICC TNM classification (the 8th edition), ¶: the Japanese Classification of Colorectal, Appendiceal, and Anal Carcinoma: the 3rd English Edition (†: *p* < 0.1 *: *p* < 0.05).

**Table 4 jcm-09-02509-t004:** Relationship between the levels of microRNAs in preoperative plasma and clinicopathological characteristics.

			Target MicroRNAs					
			MIR21		MIR29a		MIR92a	
			−∆Ct value ^‡^	*p*-value	−∆Ct value ^‡^	*p*-value	−∆Ct value^‡^	*p*-value
Gender	Male	17 (38.6)	−2.54 [−3.91–0.14)	0.63	−3.86 (−5.63–−1.3)	0.44	−1.79 (−2.95–−0.59)	0.98
	Female	27 (61.4)	−2.61 (−4.31–0.07)		−3.76 (−1.14–−6.33)		−1.70 (−2.78–−0.95)	
Age	≥75	14 (31.8)	−2.74 (−3.91–0.07)	0.22	−3.58 (−5.42–−2.32)	0.74	−1.66 (−2.36–−0.92)	0.17
	<75	30 (68.2)	−2.86 (−4.31–0.14)		−4.03 (−6.33–−1.14)		−1.85 (−2.95–−0.59)	
BMI	≥22	22 (50.0)	−2.62 (−4.31–0.14)	0.92	−4.08 (−6.33–−1.35)	0.21	−1.80 (−2.95–−0.59)	0.34
	<22	22 (40.0)	−2.57 (−3.98–−0.93)		−3.61 (−5.97–−1.14)		−1.73 (−2.36–−0.90)	
Tumor location	Colon	27 (61.4)	−2.52 (−3.91–0.07)	0.71	−3.92 (−6.33–−1.14)	0.67	−1.81 (−2.95–−0.90)	0.70
	Rectum	17 (38.6)	−2.61 (−4.31–0.14)		−3.76 (−5.97–−1.35)		−1.75 (−2.57–−0.59)	
	Right side	17 (38.6)	−2.48 (−3.91–0.07)	0.17	−3.38 (−6.13–−1.14)	0.16	−1.66 (−2.36–−0.90)	0.23
	Left side	27 (61.4)	−2.84 (−4.31–0.14)		−4.01 (−6.33–−1.35)		−1.83 (−2.95–−0.59)	
Preoperative CEA	Elevated	18 (40.1)	−2.52 (−4.31–−0.93)	0.51	−3.66 (−6.13–−1.14)	0.51	−1.78 (−2.57–−0.59)	0.74
	Normal	26 (59.1)	−2.86 (−3.98–0.14)		−3.86 (−6.33–−1.35)		−1.77 (−2.95–−0.99)	
Preoperative CA19-9	Elevated	12 (27.3)	−2.45 (−3.54–0.14)	0.07†	−3.66 (−4.76–−1.14)	0.10	−1.81 (−2.95–−0.99)	0.58
	Normal	32 (72.7)	−2.77 (−4.31–0.07)		−3.86 (−6.33–−1.35)		−1.77 (−2.95–−0.95)	
Maximum tumor diameter	≥45	22 (50.0)	−2.50 (−3.61–0.14)	0.07†	−3.68 (−5.32–−1.30)	0.046*	−1.77 (−2.95–−0.59)	0.30
	<45	22 (50.0)	−2.99 (−4.31–0.07)		−4.08 (−6.33–−1.35)		−1.70 (−2.78–−1.38)	
Pathological T stage ^§^	T3 + T4	33 (75.0)	−2.52 (−4.31–0.14)	0.14	−3.70 (−6.13–−1.14)	0.19	−1.75 (−2.95–−0.59)	0.64
	T1 + T2	11 (25.0)	−2.99 (−3.98–−1.46)		−4.03 (−6.33–−2.32)		−1.79 (−2.78–−1.39)	
Pathological N stage ^§^	Present	20 (45.4)	−2.53 (−3.97–0.14)	0.22	−3.70 (−5.56–−1.14)	0.09†	−1.77 (−2.33–−0.59)	0.39
	Absent	24 (54.5)	−2.77 (−4.31–0.07)		−4.03 (−6.33–−1.61)		−1.76 (−2.95–−0.95)	
Pathological Stage ^§^	III + IV	22 (50.0)	−2.52 (−3.97–0.14)	0.21	−3.66 (−5.56–−1.14)	0.02*	−1.77 (−2.33–−0.59)	0.24
	I + II	22 (50.0)	−2.90 (−4.31–0.07)		−4.09 (−6.33–−2.32)		−1.76 (−2.95–−1.07)	
Histological differentiation ^¶^	Poorly	6 (13.6)	−2.70 (−3.21–0.14)	0.24	−3.66 (−3.86–−1.30)	0.22	−1.98 (−2.33–−0.92)	0.56
	Well and moderately	38 (86.4)	−2.58 (−4.31–0.07)		−4.03 (−6.33–−1.14)		−1.73 (−2.95–−0.59)	
Lymphatic invasion ^¶^	High	13 (29.5)	−1.92 (−3.98–0.14)	0.05†	−3.58 (−5.97–−1.14)	0.05†	−1.75 (−2.29–−0.9)	0.61
	Mild	31 (70.4)	−2.69 (−4.31–0.07)		−4.03 (−6.33–−1.61)		−1.81 (−2.95–−0.59)	
Venous invasion ^¶^	High	15 (34.1)	−2.43 (−3.98–−1.20)	0.85	−4.14 (−5.97–−1.30)	0.43	−1.75 (−2.26–−0.59)	0.10
	Mild	29 (65.9)	−2.84 (−4.31–0.14)		−3.66 (−6.33–−1.14)		−1.81 (−2.95–−0.92)	
Perineural invasion ^¶^	Positive	25 (56.8)	−2.48 (−3.61–0.14)	0.01*	−3.66 (−6.13–−1.14)	0.26	−1.81 (−2.95–−0.59)	0.58
	Negative	19 (43.2)	−3.08 (−4.31–−1.49)		−4.03 (−6.33–−1.35)		−1.69 (−2.78–−1.39)	

CEA: Carcinoembryonic antigen level, normal upper limit at 5 ng/mL, CA19-9: Carbohydrate antigen 19-9 level, normal upper limit at 37 ng/mL, ‡: −∆Ct value = −(Ct value (target MIR)–Ct value (internal control MIR16)), §: UICC TNM classification (the 8th edition), ¶: the Japanese Classification of Colorectal, Appendiceal, and Anal Carcinoma: the 3rd English Edition (†: *p* < 0.1 *: *p* < 0.05).

**Table 5 jcm-09-02509-t005:** Multivariate analysis of microRNAs and pathological characteristics in CRC with the high lymphatic invasion.

Prognostic factors	*p*-Value	Odds Ratio	95% Confidence Interval
Pathological T stage ^§^ (T3+T4/T1+T2)	0.66	1.84	0.12–33.45
Venous invasion ^¶^ (high/mild)	<0.001 ***	66.84	5.10–2853.98
Perineural invasion ^¶^ (positive/negative)	0.53	0.46	0.03–5.70
Level of MIR92a in tumor tissue (−∆Ct value^‡^) (high/low)	0.01 *	0.08	0.003–0.60
Level of MIR21 in preoperative plasma (−∆Ct value ^‡^) (high/low)	0.19	0.20	0.008–2.16
Level of MIR29a in preoperative plasma (−∆Ct value ^‡^) (high/low)	0.009 **	30.72	2.24–1038.11

‡: −∆Ct value = −(Ct value (target MIR)–Ct value (internal control MIR16)), §: UICC TNM classification (the 8th edition), ¶: the Japanese Classification of Colorectal, Appendiceal, and Anal Carcinoma: the 3rd English Edition (*: *p* < 0.05 **: *p* < 0.01 ***: *p* < 0.001).

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
