# Peer review of "Tumor Tissue MIR92a and Plasma MIRs21 and 29a as Predictive Biomarkers Associated with Clinicopathological Features and Surgical Resection in a Prospective Study on Colorectal Cancer Patients"

_jcm, 2020, doi:10.3390/jcm9082509_

Round 1

Reviewer 1 Report

The authors present an interesting short study on the potential use of MiRNAs as biomarkers for various features of CRC. They aim to address an important clinical need for a more simplified testing protocol that decreases the need for multiple, expensive, and possibly invasive, tests. While the authors do not find a magic bullet that meets all of these goals in this early study, it is clear that they have made significant progress in advancing the goal. With that, I had a few questions about the approach and just a few comments on other structural issues with the manuscript.

General study question:

Based on the goal outline in the introduction, I was hoping to see some easily-detectable biomarker that would correlate with clinicopathology in both tissue and plasma, but the data show considerable variation. Granted that this was a small initial study, can the authors speak to plans of expanding the patient population in the currently ongoing studies? More specifically, are samples from other populations outside of Japan being considered and how might that affect the interpretation of the data in this manuscript?

Manuscript structure:

Line 52- the word study should be studies

Line 110- the word samples should be sample

Figures 2-6- The image resolution is too low and/or the font size too small. The images appear blurry on screen and in print. The line/point weight should also be increased to increase the resolution between samples represented in these graphs.

Lines 402-403- The last sentence of the conclusion is a bit overstated for the data presented. I would suggest taking out the last part stating, “a predictor of reoccurrence”. While this may be proven in the ongoing work of the authors, this manuscript doesn’t address this feature.

Author Response

Thank you for pointing out our mistakes. These points have been rewritten into proper sentences. We think your comments have allowed us to significantly improve our paper. Thank you.

Reviewer1

General study question:

・Based on the goal outline in the introduction, I was hoping to see some easily-detectable biomarker that would correlate with clinicopathology in both tissue and plasma, but the data show considerable variation.

Granted that this was a small initial study, can the authors speak to plans of expanding the patient population in the currently ongoing studies?

(answer)

In this study, we report the results of an analysis limited to 44 cases in which both blood and tissue samples could be collected. However, it has been clearly demonstrated that the plasma MIRs 21 and 29a levels are correlated to the progression of stage and lymphatic invasion.

・More specifically, are samples from other populations outside of Japan being considered and how might that affect the interpretation of the data in this manuscript?

(answer)

Only the patients who have undergone resection of colorectal cancer at Gifu University Hospital are only covered. In the future, we are planning a prospective clinical trial in collaboration with other institutions, but it seems that it will be limited to cases in Japan. However, I do not think that the difference in epidemiological or pathological features of colorectal cancers between in Japan and in outside of Japan is so important as to overturn the results of this study, so I do not think that it will have a significant impact.

・Line 52- the word study should be studies

(answer)

Thank you for pointing out. We have corrected the relevant part.

・Line 110- the word samples should be sample

(answer)

Thank you for pointing out. We have corrected the relevant part.

・Figures 2-6- The image resolution is too low and/or the font size too small. The images appear blurry on screen and in print. The line/point weight should also be increased to increase the resolution between samples represented in these graphs.

(answer)

Thank you for pointing out. The figures have been modified for clarity.

・Lines 402-403- The last sentence of the conclusion is a bit overstated for the data presented. I would suggest taking out the last part stating, “a predictor of reoccurrence”. While this may be proven in the ongoing work of the authors, this manuscript doesn’t address this feature.

(answer)

Thank you for pointing out. In this study, the last sentence of the conclusion is a bit overstated, so the relevant parts have been deleted.

In closing, let me thank you once again for your extremely cogent comments which have helped us improve the quality of our paper.

Reviewer 2 Report

The work is interesting because it provides information about application of tumour tissue MIR92a and plasma MIR21s and MIR29a as a potential predictive biomarkers in CRC patients in relation to clinicopathological features and surgical resection.

The paper is well written, and majority of the sections are clear and exhaustive. Thus I recommend the work for publication provided that some points are addressed.

Abstract

Line 13 – high lymphatic invasion? What does it mean. I didn’t find any information about this classification in Material and methods and I suggest to describe it in this section.

Line 16 – in cases with high plasma miRNA level – did you mean total miRNA level?

Introduction:

Line 50 – should be apoptotic instead of apototic

Material and methods:

I have reservations regarding the section 2.6 Collection of clinical and pathological characteristics.

Line 154 – should be the 3rd English Edition

I don’t understand classifications and abbreviations used in your work.

Line 156 - well and moderately differentiated group (tub1,tub2) vs. poorly differentiated group (por, muc). Why do you divide poorly differentiated tumour into two groups – por and muc and what does it mean? Similarly in the case of - well and moderately differentiated – tub1, tub2???. I suggest to describe and explain their abbreviations and definitions.

The classification description scheme needs improvement. You should use a full names of analyzed parameters in section Material and methods, which should introduce the readers and show a detailed methods used in work. You divide patients with lymphatic invasion into two groups: mild: Ly0 – I guess that it means patients without lymphatic invasion? What does mean Ly1a, Ly1b, Ly1c??? You should describe it. Similarly, line 157 – V – venous invasion. V0 – patients without venous invasion but V1a, V1b, V1c, Pn1a, Pn1b? It would be good to write the explanation of letters T, N and M of TNM classification and change the names in the all tables (e.g. depth of tumour invasion, presence of lymph node or distant metastasis),

Results:

Table 1 – What mean symbols C3, A8, T6 etc. in description of primary colorectal tumor location. It is not clearly. You should explain these abbreviations below the table.

It would be interesting to include sentence, information about part of colon belong to the right and left side briefly for the benefits of general readership.

Macroscopic types – I don’t know what mean symbols Type 0, Type 1, Type 2 and Type5 – lack of information about it in material and methods.

I suggest to place the information about CEA and CA 19-9 levels at the end of the table.

Line 181 – Why did you evaluate the relative levels of exosomal miRNAs to circulating miRNAs only  in 4 preoperative plasma samples?

Line 212 – You should change the phrase stage>III into III+IV (in all text and all tables).

Table 3

I suggest to change phrase pathological T stage into depth of tumour invasion. What mean >3 and <3? T3+T4 and T1+T2? If yes, change it. Pathological N stage change into presence of lymph node metastasis and division into present or absent. Change pathological stage >3 and <3 into III+IV and I+II.

 Figure 3 - on the figures are shown correlations or comparison of groups? Headings may be misleading.

Change the description of pT <3, >3 (T1+T2 and T3+T4) and in the same way pStage.

Line 230 – Was the MIR92a the weakest? P-value and r-coefficient provide that MIR21 was the weakest, wasn’t it?

Please, add one sentence about relationship between MIR29a and presence of lymph node metastasis (p=0.02) in the section 3.6. Relationship between levels of miRNAs in preoperative plasma and clinicopathological characteristics.

Is the figure 5 reflect the values from the table 4? If yes I see different p-values in case of relationship between MIR29a and pN stage. In the table p=0.09 but on the figure NS.

You should explain the abbreviations Pre, POD7 and POM1 below the figure 6.

Line 309 - may rapidly reflect the amount of cancer. Do you mean residual amount of cancer?

Author Response

Thank you for pointing out our mistakes. These points have been rewritten into proper sentences. We think your comments have allowed us to significantly improve our paper. Thank you.

Reviewer2

Abstract

・Line 13 – high lymphatic invasion? What does it mean. I didn’t find any information about this classification in Material and methods and I suggest to describe it in this section.

(answer)

Thank you for pointing out. I have added a detailed explanation to the sections on materials and method, including lymphatic invasion, venous invasion, perineural invasion, differentiation, and macroscopic classification.

・Line 16 – in cases with high plasma miRNA level – did you mean total miRNA level?

(answer)

Yes. This "plasma miRNA" means total miRNA level.

Introduction:

・Line 50 – should be apoptotic instead of apototic

(answer)

Thank you for pointing out. We have corrected the relevant part.

Material and methods:

・Line 154 – should be the 3rd English Edition

(answer)

Thank you for pointing out. We have corrected the relevant part.

・Line 156 - well and moderately differentiated group (tub1,tub2) vs. poorly differentiated group (por, muc). Why do you divide poorly differentiated tumour into two groups – por and muc and what does it mean? Similarly in the case of - well and moderately differentiated – tub1, tub2???. I suggest to describe and explain their abbreviations and definitions.

The classification description scheme needs improvement. You should use a full names of analyzed parameters in section Material and methods, which should introduce the readers and show a detailed methods used in work. You divide patients with lymphatic invasion into two groups: mild: Ly0 – I guess that it means patients without lymphatic invasion? What does mean Ly1a, Ly1b, Ly1c??? You should describe it. Similarly, line 157 – V – venous invasion. V0 – patients without venous invasion but V1a, V1b, V1c, Pn1a, Pn1b? It would be good to write the explanation of letters T, N and M of TNM classification and change the names in the all tables (e.g. depth of tumour invasion, presence of lymph node or distant metastasis),

(answer)

Thank you for pointing out. In the materials and methods section, we have added a detailed explanation on the degree of differentiation, lymphatic invasion, venous invasion, perineural invasion, and macroscopic classification of tumors. At the same time, we have added a description of TNM classification.

Results:

・Table 1 – What mean symbols C3, A8, T6 etc. in description of primary colorectal tumor location. It is not clearly. You should explain these abbreviations below the table.

(answer)

Thank you for pointing out. I have added a description.

・It would be interesting to include sentence, information about part of colon belong to the right and left side briefly for the benefits of general readership.

(answer)

Thank you for pointing out. I added the explanation to the table.

・Macroscopic types – I don’t know what mean symbols Type 0, Type 1, Type 2 and Type5 – lack of information about it in material and methods.

(answer)

Thank you for pointing out. Added description in materials and method section.

・I suggest to place the information about CEA and CA 19-9 levels at the end of the table.

(answer)

Thank you for pointing out. We have corrected the relevant part.

・Line 181 – Why did you evaluate the relative levels of exosomal miRNAs to circulating miRNAs only  in 4 preoperative plasma samples?

(answer)

To confirm that the proportion of exosomal miRNA in circulating miRNA (total plasma miRNA) is extremely small, we first compared the expression levels with a small number of samples. As a result, the number of samples was small, but the results were similar in all cases (the relative expression level of exosomal miRNA was very small and less than 5%). Therefore, we decided that it is not necessary to increase the number of samples any more.

・Line 212 – You should change the phrase stage>III into III+IV (in all text and all tables).

(answer)

Thank you for pointing out. We have corrected the relevant part.

Table 3

・I suggest to change phrase pathological T stage into depth of tumor invasion. What mean >3 and <3? T3+T4 and T1+T2? If yes, change it. Pathological N stage change into presence of lymph node metastasis and division into present or absent. Change pathological stage >3 and <3 into III+IV and I+II.

(answer)

Thank you for pointing out. We have corrected the relevant part.

・ Figure 3 - on the figures are shown correlations or comparison of groups? Headings may be misleading.

(answer)

Thank you for pointing out. We have corrected the relevant part.

・Change the description of pT <3, >3 (T1+T2 and T3+T4) and in the same way pStage.

(answer)

Thank you for pointing out. We have corrected the relevant part.

・Line 230 – Was the MIR92a the weakest? P-value and r-coefficient provide that MIR21 was the weakest, wasn’t it?

(answer)

Since no significant correlation (p<0.05) was found in any MIRs, it seems that the correlation coefficient has little statistical significance. Focusing on the distribution of the graph rather than that, MIR92 is clearly distributed almost parallel to the x-axis compared to other MIRs, and it is considered that the correlation between expression in tissue and expression in plasma is weakest. As a result, MIR92a showed no correlation with plasma expression and clinicopathological characteristics.

・Please, add one sentence about relationship between MIR29a and presence of lymph node metastasis (p=0.02) in the section 3.6. Relationship between levels of miRNAs in preoperative plasma and clinicopathological characteristics.

(answer)

The p-value for MIR29a and lymph node metastasis is 0.09, showing a significant tendency (p<0.1), but no significant difference (p<0.05).

・Is the figure 5 reflect the values from the table 4? If yes I see different p-values in case of relationship between MIR29a and pN stage. In the table p=0.09 but on the figure NS.

(answer)

Thank you for pointing out. We have corrected the relevant part.

・You should explain the abbreviations Pre, POD7 and POM1 below the figure 6.

(answer)

Thank you for pointing out. I added the explanation of the abbreviation.

・Line 309 - may rapidly reflect the amount of cancer. Do you mean residual amount of cancer?

(answer)

Thank you for pointing out. We have corrected the relevant part.

In closing, let me thank you once again for your extremely cogent comments which have helped us improve the quality of our paper.